# Description of an activity-based enzyme biosensor for lung cancer detection
Paul W. Dempsey [1] ✉, Cristina-Mihaela Sandu[1], Ricardo Gonzalezirias[1], Spencer Hantula[1],
Obdulia Covarrubias-Zambrano[2], Stefan H. Bossmann [2], Alykhan S. Nagji[2], Nirmal K. Veeramachaneni[3],
Nezih O. Ermerak[4], Derya Kocakaya [4], Tunc Lacin[4], Bedrittin Yildizeli[4], Patrick Lilley[5], Sara W. C. Wen [6],
Line Nederby[6], Torben F. Hansen[6] & Ole Hilberg[6]

## Abstract

**Background** Lung cancer is associated with the greatest cancer mortality as it typically presents with incurable distributed disease. Biomarkers relevant to risk assessment for the detection of lung cancer continue to be a challenge because they are often not detectable during the asymptomatic curable stage of the disease. A solution to population-scale testing for lung cancer will require a combination of performance, scalability, cost-effectiveness, and simplicity.

**Methods** One solution is to measure the activity of serum available enzymes that contribute to the transformation process rather than counting biomarkers. Protease enzymes modify the environment during tumor growth and present an attractive target for detection. An activity based sensor platform sensitive to active protease enzymes is presented. A panel of 18 sensors was used to measure 750 sera samples from participants at increased risk for lung cancer with or without the disease.

**Results** A machine learning approach is applied to generate algorithms that detect 90% of cancer patients overall with a specificity of 82% including 90% sensitivity in Stage I when disease intervention is most effective and detection more challenging.

**Conclusion** This approach is promising as a scalable, clinically useful platform to help detect patients who have lung cancer using a simple blood sample. The performance and cost profile is being pursued in studies as a platform for population wide screening.

## Plain language summary

Lung cancer is responsible for more deaths worldwide than all other cancers. It is often detected with the appearance of symptoms when treatment is limited and outcomes for the patient are much worse. While imaging chest scans can detect disease, they are poorly used even in the United States where it is an approved screening method. When cancer is present, protease enzymes are responsible for making space and modifying the lung tissue for the growing tumor. This report describes a panel of 18 sensors that release a fluorescent signal when these enzymes are present in a blood sample. The signal acts like a fingerprint of activity that can be used to identify people with lung cancer. This sensor platform can detect patients with curable lung cancer and could provide a platform for screening very large populations of at-risk individuals.

Lung cancer is associated with the greatest mortality of all cancers worldwide. In the United States (US), over 230,000 new cases are reported annually. Once the disease is advanced, the 5-year survival rates are 6%[1]. The National Lung Cancer Screening Trial (NLST), a large prospective randomized screening trial in the US demonstrated a 20% decrease in mortality using chest low dose computed tomography (LDCT)[2]. Based on the US Preventative Services Task Force (USPSTF) guidelines, there are 15.5 M individuals in the US who require screening for lung cancer based on age and smoking related risk factors[3]. However, less than 5% of this at risk population was successfully screened in 2015[4]. Compounding this compliance problem, LDCT screening methods present with large numbers of false positive screens. The NLST initially reported a 73.4% specificity highlighting the challenge of identifying lung cancer by imaging alone[5]. The burdensome 96% false positive rate was addressed by instituting Lung-RADS classification criteria to standardize analysis in the US. This improved baseline specificity to 87.2% with an attendant decrease in sensitivity to 84.9%[6]. So there remains a serious unmet clinical need for additional tools to identify lung cancer in the large at-risk population across all stages of the disease process.

Technological improvements in imaging, next generation sequencing and protein quantitation have added many more data points to the search space for clinically relevant biomarkers. Despite this, there has been little

[1]Hawkeye Bio, Inc, Torrance, CA, USA. [2]University of Kansas Medical Center (KUMC), Kansas City, KS, USA. [3]St. Louis University, School of Medicine, St Louis, MO, USA. [4]Marmara University, Istanbul, Turkey. [5]Liquid Biosciences, Inc, Aliso Viejo, CA, USA. [6]Vejle Hospital, University Hospital of Southern Denmark, Vejle, Denmark. ✉e-mail: pdempsey@hawkeyebio.com

overall improvement in patient outcomes for several reasons: It is during the earliest stage of lung cancer that intervention results in a significant reduction in mortality[7]. Confounding this, there are few examples of biomarkers that are present in sufficient quantities to be specifically detected in the earliest stages of cancer when overall survival can be successfully influenced[8] and compliance for lung cancer screening programs has been challenging[9].

Activity based sensors (ABS) rely on measuring enzyme activity from the biomarker rather than enumeration[10]. Protease enzymes are particularly good targets for ABS as they are a large family of enzymes involved in many biological processes and they have a well-defined role in cancer. Being irreversible modifiers, proteases are carefully regulated with impact on tumor cell invasion, angiogenesis, epithelial to mesenchymal transition, and ultimately malignancy[11,12]. We hypothesize that a panel of protease targets selective for a range of protease enzymes can build a "fingerprint" of activity characteristic of the disease process associated with lung cancer. This approach, termed Lung Enzyme Activity Profile (LEAP), is comprised of a panel of ABS that responds to the presence of activated protease enzymes in a blood sample selected by a machine learning approach to classify signal distinguishing patients with or without lung cancer. The nature of the assay makes this technique a possible solution for population scale testing unrestricted by traditional risk definitions.

## Methods

### Manufacture of sensors

A solution of 2.0 g graphene (HydroGraph FGA 0.3, produced under GMP conditions) was dissolved in 20 ml dimethylformamide (DMF, Fisher Scientific, Hampton NH). The solution was then dispersed using a probe sonicator (Fisherbrand FB705 Sonic Dismemberator) using 20 s on and 10 s off pulses at intensity level 35 for 60 min. The dispersed graphene was converted to carboxygraphene (CG) by warming 2.0 g dispersed graphene in 40 ml DMF in a 250 ml Erlenmeyer flask suspended in a room temperature silicone oil bath with a magnetic stir bar. The temperature of the reaction was raised to 40 °C and 1.0 g of 5-bromovaleric acid (Sigma-Aldrich) was added to the graphene suspension followed by slow addition of 0.36 g sodium azide crystals (Fisher Scientific). Once all the reagents were dissolved, the temperature was ramped linearly at 1 °C per minute to 75 °C, incubated for 1 h and then allowed to cool to room temperature. The suspension was washed 3-times by centrifugation at $10,000 \times g$ for 5 min with 100% ethyl alcohol (Sigma Aldrich) and once with DMF before storage in DMF.

For manufacture of the polyethylenimine (PEI)-derivatized carboxygraphene (CGP) intermediate, 1.0 g of CG was dispersed in 30 ml DMF in a flask with a stir bar. 0.54 g 1-ethyl-3-(3-dimethylaminopropyl)-carbodiimide (EDC) and 0.53 g 4-dimethylaminopyridine (DMAP) were added and stirred until dissolved at room temperature. 1.2 g PEI (10,000 MW branched, Sigma-Aldrich) was added and the reaction stirred for 2 h. The CGP particles were washed 3-times with ethyl alcohol using centrifugation as above and once with DMF. The CGP particles were stored in DMF.

The last step is the conjugation of the sensor peptide-tetrakis-carboxyphenyl-porphyrin (TCPP). 103 mg of CGP was dispersed in 16 ml DMF using a sonicating water bath for 1 min. 10.5 mg of EDC and 11 mg of DMAP was added to the CGP with stirring at room temperature. 7.0 mg of TCPP labeled peptide (Aapptec, Louisville KY) was then added and the reaction stirred for 2 h at room temperature. The resulting biosensor was washed 3-times with ethyl alcohol and once with ethyl ether (Fisher Scientific) before being dried at 40 °C under vacuum. The final biosensor product was stored at −20 °C under argon.

### DLS and Zeta potential measurements

Dynamic light scattering (DLS) and Zeta potential measurements were taken using a Malvern Instruments Zetasizer. For this, a 10 µg/ml solution of sensor at each stage of manufacture was prepared using highly pure water for Zeta potential (>18 MΩ-cm HPLC water, Fisher Cat#: W7-4) or 2-(N-morpholino)ethanesulfonic acid (MES) buffer at pH 6 for DLS.

### Biosensor assay

For biosensor measurements using wet chemistry, solutions of graphene nanosensors were prepared as working solutions at 25.0 µg/ml in 10 mM MES pH 7.4 containing 155 mM NaCl and 10 µM final concentrations each of $MgCl_2$, $CaCl_2$ and $ZnCl_2$. The working solutions were sonicated for 10 min in a sonicating water bath. 125.0 µl biosensor working solution was dispensed in replicates in a black 96 well plate. 5.0 µl serum or control was added to each well and the plate was incubated for incremental periods of time in a VarioSkan Lux plate reader (Thermo Fisher). Fluorescence was measured with 422 nm ± 5 nm excitation and 650 nm ± 12 nm emission at defined intervals throughout the experiment. The chamber temperature was maintained at 37, 41, or 45 °C as indicated.

For biosensor stimulations, 4x working solutions were prepared for each biosensor (100 µg/ml) in 40 mM MES containing 40 µM final concentrations each of $MgCl_2$, $CaCl_2$ and $ZnCl_2$ and sonicated. Duplicate 20 µl aliquots of all 18 biosensors supplemented with excipient were dispensed in an array on a 384-well black plate and lyophilized. Plates were heat sealed and stored at room temperature in mylar bags with desiccant until used. For use, each well was resuspended with 30.0 µL 0.9% NaCl solution for 2 min at room temperature. Then 8.0 µl of sera diluted into 42.0 µl of 0.9% NaCl was added to each well in a panel of replicated biosensor wells for an 80.0 µL and 1:10 sera diluted final volume. Samples were mixed by pipetting and the plates were incubated at the indicated temperature in the VarioSkan Lux plate reader for 60 min. Fluorescence was measured with 422 nm ± 5 nm excitation and 650 nm ± 12 nm emission at 10-min intervals throughout the experiment.

### Serum samples

Blood samples from volunteers with confirmed absence of lung cancer or presence of pathologically confirmed untreated lung cancer were collected at The University of Kansas Medical Center (KUMC)(Human Research Protection Program IRB#: STUDY00144465), Marmara University (Istanbul, Turkey)(Clinical Research Ethics Committee Decision number 898 and the Turkish Medicines and Medical Devices Agency E-68869993-511.06-570065), and the University of Southern Denmark (Vejle, Denmark)(University of Southern Denmark Ethics Committee IRB S-2022014) using a standard clinical protocol. All collections were approved by IRB at the individual institutions and informed consent was obtained from each patient in accordance with the Declaration of Helsinki. Blood samples were collected in a BD Serum Separator Tube (BD Cat # 367981). After collection, the blood tube was maintained in an upright position for 30–60 min at room temperature before being centrifuged for 10 min at 1300 r.c.f. After serum separation, sera were recovered using a transfer pipette to freezer vials which were placed at −80 °C within 4 h of sample harvest.

### Machine learning analysis

We use Emerge, a quantitative artificial machine learning tools set (Liquid Biosciences, Inc., Aliso Viejo CA). Briefly, the software has seven key features:

i.   Candidate algorithms are randomly assembled from a variety of mathematical primordia, including variables and functions, such as trigonometric functions (e.g., c + sine x, where c is a constant and x is a variable).

ii.  Candidate algorithms are placed within $S$ subpopulations of size $n_i$, with a total population size of sigma $n_i$.

iii. At the start of each generation, from each subpopulation $n_i/2$ pairs are randomly chosen. From each pair, the algorithm with the greatest fitness is chosen to survive, the other being discarded. Fitness is determined according to the formal nature of the algorithm task. That is classification accuracy in the case of this classification application.

iv.  Surviving algorithms either make a copy of themselves, where that copy is mutated, or they undergo recombination with other surviving algorithms to create two new offspring. The parental algorithms are retained in the population in both cases.

**Fig. 1 | Assembly and activation of biosensors.** Turbostratic graphene was deagglomerated using sonication (**A**) and subsequently modified by carboxylation using bromovaleric acid (**B**). The sensors are further modified by coating with a polymer layer of polyethylenimine (PEI) (**C**). Enzyme specific peptide sequences terminated with tetrakis-carboxyphenyl-porphorin (TCPP) are added to the surface (**D**). Incubation with serum containing active protease enzyme results in cleavage of the cognate peptide allowing fluorescence detection in a fluorescent plate reader.

v. After an initial period of $w$ generations, every $g$ generations the best algorithms within a subpopulation (e.g., the best $p$ percent) migrate to their nearest neighbor subpopulation in a toroid pattern with a unidirectional flow. Such migration is without replacement.

vi. After $w + gS$ generations, the remaining algorithms are evaluated on the Selection data subset with respect to both accuracy and reliability. Accuracy is reflected by a combination of sensitivity and specificity as measured in the Selection subset. Reliability measures consistency of this Accuracy between training and selection data subsets. A single algorithm is automatically selected based on a combination of these two measures. The accuracy of this algorithm is then measured in the out-of-sample Test data subset, not for any further selection but to provide an estimate of performance on future prospective samples from patients under the same assay process, and under approximately the same patient selection criteria.

vii. A third subset of data is used to test the resultant winning algorithm with respect to its fitness and consistency. The results of this final test are taken to provide an estimate of algorithm performance on subsequent novel bodies of data.

This methodology uses predictive or classification fitness in an evolutionary setting to evolve by mutation, cross-over or migration, and then naturally select Turing machines selecting for improved algorithmic progeny with each generation. By evaluating on the order of $10^{15}$ algorithms, Emerge effectively replicates natural selection in silico using Turing machines to capture biological aspects of the relationship between the biosensor activity and the presence or absence of disease. Each algorithm is comprised of a sequence of instructions. Each instruction applies a single mathematical or logical function to one or more operands, then stores the resulting value in a memory register that is used by at least one subsequent instruction. The Emerge platform has a broad palette of 48 mathematical and logical functions that are available to the evolutionary process. This approach of evolutionary computing emerged from John Holland and colleagues[13]. While neural networks, deep learning and other forms of machine learning and artificial intelligence are frequently cited, evolutionary computing methods are emerging as superior to such mainstream technologies in complex quantitative problem domains characterized as inverse problems[14–16].

### Reporting summary
Further information on research design is available in the Nature Portfolio Reporting Summary linked to this article.

## Results
### Graphene biosensor assembly
Detection of protease enzymes has traditionally relied on expression systems such as ELISA based affinity methods[17]. Numerous approaches to measuring the endogenous activity of protease enzymes using more sensitive fluorescence based activity assays have been developed[18,19]. However, the ability to detect endogenous protease activity in a complex medium such as serum has been more complicated[20]. While graphite derivatives are attractive as efficient broad spectrum quenchers of fluorescent dyes, the stringent chemical modification required to convert graphite to graphene is difficult to standardize with the product producing varied noise backgrounds[21].

We have developed a biosensor using a graphene based backbone that relies on the reproducible structure of explosion-synthesized turbostratic graphene[22]. The Raman spectra of the core particle show characteristic graphene features with $D$, $G$, and $2D$ peaks centered at 1340, 1575, and 2680 cm$^{-1}$ respectively (Supplementary Fig. 1). For particle assembly, explosion-synthesized graphene was prepared by deagglomeration of turbostratic fractals (HydroGraph, Toronto, Canada) using sonication. The deagglomerated few-layer graphene $n = 7.2 +/- 2.3$ layers, (Supplementary Table 1 and Supplementary Fig. 2) was surface-carboxylated and then coated with a water soluble polyethyleneimine (PEI) skin. The PEI-modified carboxygraphene was further functionalized with peptide labeled with the fluorescent dye tetrakis-carboxyphenyl-porphyrin (TCPP) (Aapptec, Louisville, KY) (Fig. 1). The biosensor particle size, by DLS, averaged 473 nm (s.d. = 77 nm) in the initial graphene particles to 239 nm (s.d. = 40 nm) in the completed biosensor (Supplementary Table 2). The particle size was also measured using a microfluidic resistive pulse sensing Spectradyne nCS1 platform. In agreement with the DLS observations, the weighted average of the biosensor particles measured between 87 and 212 nm in diameter (Supplementary Fig. 3). The Zeta potential surface charge decreased as the surface was modified first with carboxylic acid (−14 mV average Zeta potential) followed by PEI tethering (41 mV average) and then the addition of the dye-peptide moiety (31 mV) (Supplementary Table 2) consistent with the additional charge added to the surface. Furthermore, elemental analysis of the biosensors during fabrication showed incremental inclusion of Hydrogen and Oxygen after carboxylation of the graphene backbone. Addition of Nitrogen was observed only after addition of the PEI skin (Supplementary Table 3). The biosensors so produced maintained stable colloidal solutions for up to nine months (Supplementary Table 4).

To design peptide sequences for selection of protease activity, we reasoned that protease enzymes, which demonstrated significantly altered transcriptional expression between primary tumor samples and healthy human tissue would be productive candidates for differential activity[23]. Both Non-Small Cell Lung Cancer (NSCLC) and Small Cell Lung Cancer (SCLC) expression datasets were interrogated and compared to matched normal lung expression to identify proteases differentially regulated in lung cancer tissue[24]. Ten proteases were identified with an expression pattern that could support differentiation of normal and lung cancer tissue ($p \leq 0.001$ for all 10 markers) as well as distinguishing SCLC and NSCLC (Supplementary Fig. 4). This list was expanded to include peptides designed in previous studies which had shown different protease enzyme profiles in breast, pancreatic and lung cancers (Table 1)[18,25–27]. We also manufactured a sensor

https://doi.org/10.1038/s43856-024-00461-7   **Article**

## Table 1 | Amino acid sequence of enzyme sensors

| Biomarker No. | Dye - Oligopeptide sequence | aa | 1° Target |
|---|---|---|---|
| BM01 | TCPP-GAGVPMSMRGGAG | 13 | MMP-1 |
| BM02 | TCPP-GAGIPVSLRSGAG | 13 | MMP-2 |
| BM03 | TCPP-GAGRPFSMIMGAG | 13 | MMP-3 |
| BM04 | TCPP-GAGVPLSLTMGAG | 13 | MMP-7 |
| BM05 | TCPP-GAGVPLSLYSGAG | 13 | MMP-9 |
| BM06 | TCPP-GAGPSGLQTGAG | 12 | MMP-10 |
| BM07 | TCPP-GAGGAANLVRGAG | 13 | MMP-11 |
| BM08 | TCPP-GAGPKALGLAAG | 12 | MMP-12 |
| BM09 | TCPP-GAGPQGLAGQRGIVAG | 16 | MMP-13 |
| BM10 | TCPP-GAESDASQTGAG | 12 | MMP-15 |
| BM11 | TCPP-GAGSLLKSRMVPNFNAG | 17 | CTS-B |
| BM12 | TCPP-GAGDSGLGRAG | 11 | CTS-D |
| BM13 | TCPP-GAGEVALVALKAG | 13 | CTS-E |
| BM14 | TCPP-GAQFVRSPSGAG | 12 | CTS-H |
| BM15 | TCPP-GAGAKLKAENNAG | 13 | CTS-K |
| BM17 | TCPP-GAGGEPVSGLPAG | 13 | NE |
| BM18 | TCPP-GAGSGRSAG | 9 | uPA |
| BM19 | TCPP-GAGRRRRRRRAG | 12 | Arginase |

## Table 2 | Demographics of participants

| Individual Donors | n = 450 |
|---|---|
| Sex, No. (%) | |
| Male | 282 (63%) |
| Female | 168 (37%) |
| Age, No. (%) | |
| 0–49 | 14 (3%) |
| 50–59 | 170 (38%) |
| 60–69 | 153 (34%) |
| 70–79 | 97 (22%) |
| ≥80 | 16 (4%) |
| Smoking Status, No. (%) | |
| Current | 238 (53%) |
| Former | 172 (38%) |
| Never | 40 (9%) |
| Site, No. (%) | |
| KUMC | 84 (19%) |
| Marmara University | 235 (52%) |
| Vejle Hospital | 131 (29%) |
| Diagnosis, No. (%) | |
| Non-Lung cancer | 318 (71%) |
| Lung Cancer | 132 (29%) |
| *Histology*, No. (%) | |
| Adenocarcinoma | 78 (59%) |
| Squamous Cell | 41 (31%) |
| Other | 6 (5%) |
| Small Cell | 1 (1%) |
| Large Cell | 1 (1%) |
| Large Cell Neuro | 1 (1%) |
| Typical Carcinoid | 2 (2%) |
| Poorly Differentiated | 1 (1%) |
| Unknown | 1 (1%) |
| *Cancer Stage*, No. (%) | |
| I | 53 (40%) |
| II | 23 (17%) |
| III | 27 (20%) |
| IV | 28 (19%) |
| Unknown | 4 (3%) |

for Arginase activity which responds to the post-translational deamination of arginine to ornithine[28] and contributes to regulation of anti-tumor inflammatory responses[29,30].

### Biosensor mode of action

The biosensors are activated when the peptide tethering TCPP to the graphene particle is severed or altered by a proteolytically active enzyme or post-translational modification, thus removing the quenching action of the graphene particle. This panel of 18 different biosensors were lyophilized into a 384-well format (Argonaut Manufacturing Services, Carlsbad, CA). For activation, the sensors were stimulated with a 1:10 dilution of sera in activation buffer with an 80 µl reaction volume (see online materials and methods). Unexpectedly, increased protease activity was observed when the assay was incubated at increased temperature. At 60 min Biomarker-12 (BM12) produces 3.0-fold more fluorescence at 45 °C than 37 °C (Student's $t$ test $p = 0.002$). At 41 °C, the increment is 1.6-fold ($p = 0.004$). The increase was 3.4-fold ($p = 0.009$) and 1.8-fold ($p = 0.002$) for BM08 (Supplementary Fig. 5).

A standard curve was included by diluting BM19 peptide-TCPP from 800 ng/ml to 25 ng/ml. The standard curve $R^2$ was >0.9999 across 85 plates (Supplementary Fig. 6, Supplementary Table 5). As each biosensor releases a peptide-TCPP fragment when activated, the standard curve was able to calculate the TCPP released for all biosensors with a lower limit of quantitation of 2.98 ng/ml. Using this approach, the range of peptide-TCPP produced by each biosensor incremented kinetically relative to the background fluorescence of diluted sera only (Supplementary Fig. 7). In the absence of serum, the sensors displayed no significant release of TCPP-peptide in 60 min (Supplementary Fig. 8A). Sera presented the greatest signal for all biosensors consistent with the elevated proteolytic profile observed in serum rather than plasma (Supplementary Fig. 8)[31]. The intra assay variation on triplicate determinations of 14 sera displayed an average coefficient of variation (CV) of 6.0% across the panel (Supplementary Table 6).

### Clinical samples

Samples were collected at two sites (KUMC and Marmara University) in prospective cohorts from participants who were defined by the USPSTF criteria as at increased risk for lung cancer based on age (between 50 and

80 years of age) and smoking history (current smokers with greater than 20 pack years or cessation within the last 15 years). Participants without lung cancer underwent chest CT screening and necessary follow up review to confirm negative status. Patients with lung cancer all had pathologically confirmed disease and were treatment naïve. A third site (Vejle Hospital) collected prospective samples from donors identified clinically as at risk for lung cancer so included some number outside the USPSTF age and smoking limitations. Patients at Vejle Hospital were followed to diagnostic resolution as negative for lung cancer by chest CT scans and necessary follow up to confirm the absence of disease. Patients positive for lung cancer all had disease pathologically confirmed (Table 2). 450 unique donor samples were evaluated using the LEAP panel. In addition, a subset of 150 samples reflecting the same distribution of site and cancer, were analyzed an additional 2 times to estimate precision for a total of 750 assays. Samples were randomized and blinded such that samples with lung cancer and from each site were distributed evenly. The fluorescence was converted to concentration and wells

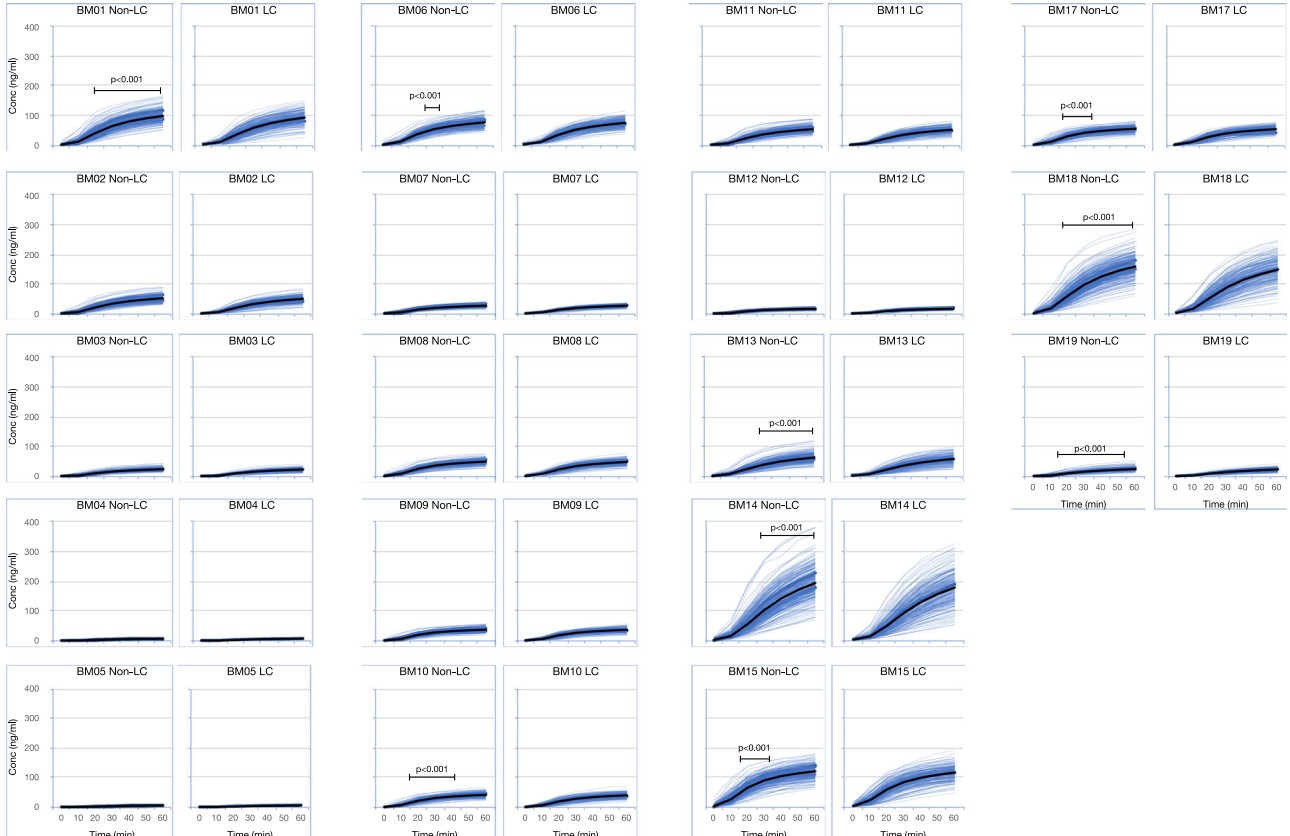

**Fig. 2 | Difference in sensor activity between patients with and without lung cancer.** The average biosensor signal at each timepoints for patients with lung cancer (LC) or without lung cancer (Non-LC) is displayed for each biomarker. For clarity, only up to 200 samples are displayed on each graph. The average signal for each group is marked with a black line. Timepoints for each biomarker which display a significant difference using a *t* test are indicated using bars annotated with the relevant P-value.

producing fluorescence below the lower limit of quantitation were marked as zero.

Analysis of overall biosensor activity revealed an average decrease in sensor activity in the context of lung cancer of 5.4% calculated as the difference between cases and controls expressed as a percent of lung cancer signal. These differences were revealed across the kinetics of the assay starting at 10 min and continuing to 60 min. Of the 108 measurements taken between 10 and 60 min, 49% demonstrated a $p \leq 0.01$ (Fig. 2). Individual timepoints for single biosensors that demonstrated the most significant differences between lung cancer and non-lung cancer were examined. Biosensors with timepoints (Biosensor_time) BM01_40, BM10_40, BM13_50, BM14_50, BM17_20, BM18_30 and BM19_20 averaged a 9.5% (range 7.7% to 12.3%) decrease in activity between lung cancer and non-lung cancer samples. The Hotelling's $T^2$ test for these measurements returned a significant result ($p = 3.38 \times 10^{-5}$) for the difference between the lung cancer and non-lung cancer cohort.

To evaluate the diagnostic capacity of this platform, analysis was performed using Emerge software. Emerge is a quantitative artificial intelligence platform designed as an unbiased methodology to produce transparent algorithms from complex biological data without prior assumptions. Biological functions result from complex networks of molecular interactions[32]. Identifying explanatory factors associated with outcomes from a set of observations in such molecular data is an "inverse" problem[33]. Traditional analytical approaches and mainstream artificial intelligence technologies are not suitable for addressing inverse problems from datasets with large numbers of variables because of the high dimensionality and non-linear relationships. Emerge was designed as a fusion of evolutionary principles, signal processing functions, and information theory to address inverse problems.

The software is agnostic to the nature of a problem in terms of explanatory variables, dimensionality or underlying mathematical relationships. Rather, it identifies both key variables and mathematical relationships associated with outcomes of interest. The data were divided into distinct, non-overlapping random subsets that were sequentially processed: A Training set, a Selection set, and a Test set segregating a third of the samples balanced for disease status, disease stage, trial site, sex, patient age, smoking status, and histology so all are represented approximately equally in each set. Analysis of the Training set provided the initial algorithmic models which were randomly generated using biomarker variables and mathematical or logical functions selected from a palette of 48 functions. These models are iteratively selected and mutated in silico to evolve models that identify relationships that support accurate classification. The resulting calculations were then evaluated on the held-out Selection subset to select a final model independent of the Training set. The performance of the final Ensemble model was confirmed on the third out-of-sample Test set. The Training, Selection, and Test data subsets were segregated as described, to avoid information leakage between discrete steps in the modeling process. As such, no resampling or cross-validation was performed to avoid overstatement of the prospective out-of-sample performance.

After the Emerge analysis, the final Ensemble applied to the Test set resulted in a 90% (95% Confidence Interval (95% CI), 80–96%) Sensitivity and 82% Specificity (95% CI, 76–88%) (Table 3). The entire data set and Test sets provided generally consistent performance results. With the application of weighted bias to the Ensemble models, the Ensemble performance was able to range from a Specificity of 86% (95% CI, 80–91%) to a Sensitivity of 97% (95% CI, 90–100%). Using the Sensitivity Max algorithm, 22 of 22 (100%) Stage IA lung cancer cases were correctly classified including 0/1 (0%), 8/9 (89%) and 11/12 (92%) of Stage IA1, IA2 and IA3 samples

**Table 3 | _Cohort Performance_: The assay performance in the entire data set (top) or the out-of-sample Test set (bottom) are indicated along with the 95% confidence intervals calculated using the Clopper-Pearson interval**

| Entire Data Set (n = 750) | | | | | |
|---|---|---|---|---|---|
| | Spec Max | Spec Bias | Ensemble | Sens Bias | Sens Max |
| Sensitivity | 61% | 63% | 84% | 85% | 96% |
| 95% CI | 55–68% | 57–70% | 79–89% | 80–89% | 92–98% |
| Specificity | 86% | 86% | 81% | 80% | 57% |
| 95% CI | 83–89% | 83–89% | 77–84% | 76–83% | 52–61% |
| NPV | 84% | 85% | 92% | 93% | 97% |
| PPV | 66% | 65% | 64% | 64% | 48% |
| Accuracy | 79% | 79% | 82% | 81% | 68% |
| 95% CI | 76–82% | 76–82% | 79–84% | 79–84% | 65–72% |
| Validation Set data (n = 250) | | | | | |
| Sensitivity | 62% | 62% | **90%** | 91% | 97% |
| 95% CI | 49–73% | 49–73% | 80–96% | 82–97% | 90–100% |
| Specificity | 86% | 85% | **82%** | 82% | 54% |
| 95% CI | 80–91% | 79–90% | 76–88% | 75–87% | 47–62% |
| NPV | 86% | 86% | 96% | 96% | 98% |
| PPV | 63% | 61% | 66% | 65% | 44% |
| Accuracy | 80% | 79% | 84% | 84% | 66% |
| 95% CI | 74–84% | 73–84% | 79–89% | 79–89% | 60–72% |

The 90% and 82% sensitivity and specificity values are bolded as they are the primary out-of-sample performance data reported in the paper.

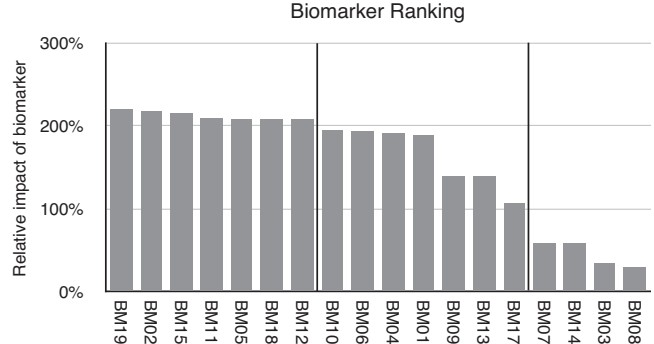

**Fig. 3 | Relative impact of biosensor panel.** Each biosensor was ranked based on the contribution of selection and diagnostic power. Results indicate performance relative to an average impact of 100%.

respectively; the setting where localized disease offers the best outcome for patients (Supplementary Table 7). The Test set demonstrated an incremental performance of 90%, 74%, 94%, and 100% Sensitivity for Stage I through IV respectively using the Ensemble. Within these subsets, the 95% CI were overlapping suggesting while LEAP maintains clinically relevant performance across all stages of disease, additional data will be needed to more clearly define performance boundaries.

Critically for the application in clinical settings, triplicate measurements of 50 sera samples in the Test set were used to estimate the precision of the system. In the Test set, a precision of >90% (Range: 90–98%) was observed for each of the algorithms (Supplementary Table 8).

The information contribution of each of the biosensors was evaluated by factoring the number of times each sensor was picked during the selection step and its contribution to the final accuracy, normalized to the average of all the metrics measured (Fig. 3). In this approach, 100% indicates a biosensor contribution equivalent to the average of the selection and diagnostic impacts. The biosensors could be separated into high (>200%), intermediate (100–200%) and low (<100%) performance relative to the average. There were 7 sensors which contributed greater than 200% relative impact suggesting there is a subset of sensors that may be sufficient when greater assay density is required. Interestingly, the sensors that demonstrated the most significant percent difference measured by difference between lung cancer and non-lung cancer cohorts were only overlapping with the sensors with greatest relative impact. The best performing time intervals for the 7 sensors that showed >200% change in relative impact produced an average decrease in signal of −7.4% (range: −2.1% to −12.3%). However, when these time intervals were tested using Hotelling's $T^2$ analysis, the p value improved to $4 \times 10^{-6}$.

This assay was evaluated here in the context of a population with a 29% cancer prevalence. In a theoretical screening population of 100,000 at-risk individuals, the NLST data predicts the prevalence of lung cancer would be 0.91%[2]. As screening compliance in the United States was 3.9% in 2015[34], a primary impediment to identifying lung cancer by screening depends on

compliance. In contrast, it has been shown that a blood test provided for the early detection of colon cancer was selected 93.5% of the time illustrating the improved compliance obtainable with a blood draw[35]. We considered the setting were the LEAP assay serves as a decision support triage tool for LDCT screening. Estimating the blood test could demonstrate 75% compliance, the LEAP triaged population would result in 520 cancers being discovered using Lung Imaging and Reporting Data System (LungRADS) assessment and follow up as practiced in the United States[6]. In contrast, with the observed 3.9% compliance, LDCT only would detect 30 lung cancers. Thus, inclusion of a modestly compliant LEAP test would increase lung cancer detection 17-fold by increasing the positive predictive value of the LDCT screen from 5.7% (95% CI, 3.9–8.1%) to 23.7% (95% CI, 21.9–25.5%) (Supplementary Table 9).

## Discussion

This study describes a blood-based ABS assay that demonstrates clinically useful detection of lung cancer across all stages. The tool relies on a fluorescent protease assay with a classification performance that would support triage of an at-risk population for lung cancer screening decision support. For the purposes of this study, the USPSTF risk criteria were used in the selection of patients. In these cohorts, we observed a sera-based specificity of 82% and sensitivity of 90% across all stages of disease. The cancer free cohort consists of age matched individuals with a similar smoking history. Moreover, because of the absence of radiation, the distributed nature of a blood sample, and the performance, the assay could be deployed in a wider risk group with the appropriate validation.

During review of the risk recommendations, the USPSTF examined 10 randomized control LDCT screening trials for accuracy and documented a mean 80.3% Sensitivity (range 59.0%–95.0%) and 76.4% Specificity (range 26.4%–99.2%)[36]. In the context of informing patient selection for LDCT screening therefore, the balanced Ensemble has a performance that could serve as a decision support tool for LDCT screening. The tunable nature of the Ensemble allows consideration in other regions outside the US that may have different selection criteria and different future implementations of LDCT screening. A bias towards Sensitivity is seen in other tests. Mathios et al., report a combination of clinical risk, CEA protein levels, and fragmentation profiles to detect 91% stage I/II patients at 80% specificity[37]. A four biomarker panel evaluated on Prostate, Lung, Colorectal, and Ovarian (PLCO) Cancer Screening Trial samples returned a Sensitivity of 85.0% and Specificity of 71.1%[38].

The participants recruited for this study used regional LDCT classification rules that will reflect the range of specificity attributable to LDCT. Without longitudinal sampling of a defined cohort, it is unclear if part of the 82% specificity observed with the LEAP assay may capture an overlap between LDCT false negative and LEAP false positive signal. Longitudinal collection of screening samples to measure LEAP sensitivity will require future clinical trials. There is also a possibility of overlap between the LEAP

profile for lung cancer with additional inflammatory diseases including other cancers. However, application to a particular risk group restricts the clinical use case to lung cancer rather than a multi-cancer detection tool.

We note that the sensors design was restricted by expression analysis of normal and lung cancer tissue. There are classes of proteases globally expressed that have clear impact on lung cancer that could be included in future panel designs to improve the performance in other patient risk groups. The Complement fragment C4d has been shown to be diagnostically elevated in bronchial fluids and saliva[39]. Analysis of peptidome profiles in lung cancer also detected degradation products of Complement C3 and C4 activation further implicating the inflammatory role of the Complement cascade[40]. The Kallikrein (KLK) family of protease, specifically KLK6, KLK13 and KLK14, have been implicated in a preclinical model of protease detectors for lung cancer[41]. These proteases may provide additional useful targets for ABS with impact on Sensitivity or Specificity[42].

Interestingly, while the biosensors were initially designed to be selective for certain proteases, the in vitro serum activity is likely selective for multiple different enzymes for each sensor. A simple estimation of the number of cleavage sites using the PROSPER tool suggests this panel is digested at 2.5 different sites per peptide on average when evaluating only 24 different proteases[43]. Therefore, as activity is revealed by the first of any active enzyme, the diagnostic power of the panel is challenging to dissect to specific enzymes.

In our analysis of the impact of each of the biosensors (Fig. 3), all the sensors were selected by Emerge to have contributed positively to the diagnostic performance. Allowing for the promiscuity of protease activity[44], the literature does support a role for many of the selected targets. Both MMP2 and its inhibitor TIMP are elevated in bronchial alveolar lavage fluid[45] and serum[46] from lung cancer patients. Cordes et al., examined the expression of all 13 Cathepsins in the context of lung cancer and found over expression of both CTSB and CTSK were significantly associated with poorer 5-year survival[47]. Previous observations by Werle et al., demonstrated significantly altered expression of CTSB and uPA and unfavorable prognosis associated with increased tissue activity of CTSB[48]. CTSH interestingly shows little change in protein expression in tissue samples but increased detection in sera maybe reflecting increased secretion, especially in smokers[49]. Arginase 1 and 2 expression is increased in lung cancer tissue[50]. Arginine depletion is suggested to alter T cell responses in vitro[50] but it is not associated with worse prognosis unless cancer associated fibroblast ARG2 expression is examined[51] possibly consistent with the boundary region expression of Arginase activity in a preclinical model[28]. Both MMP7 and MMP12 protein expression is increased in both tissue and matched serum samples[52]. MMP9 has been shown to be both over-expressed[53] and show elevated tissue activity in NSCLC tissue[54]. Conversely, ABS showed elevated MMP1 and MMP2 activity but not MMP9 activity relative to control tissue samples emphasizing the necessity of repeatable standardized measurement tools[55].

This study observed that the level of protease activity was generally decreased in lung cancer samples. This must be interpreted in the context of numerous studies describe above that measure increased expression of zymogens. That activity is decreased by 2–12% in serum samples from cancer patients suggests the cascade of zymogen activation is efficient within the proteolytic network at the tumor site, and this is reflected in a distance serum sampling event. This efficiency is not unexpected given the irreversible nature of protease activity. Alternatively, protease inhibitor ratios may be altered outside of the tumor site[53,56]. In addition, due to mutations, epimutations and misfolding events in cancer, the activity of proteases may be different than in healthy tissue. For instance, proteases which require acidic pH for activation (e.g., cathepsins), show different activity at neutral pH. Consequently, the network of proteases is disturbed, which can be detected with a panel of proteases using serum from healthy patients as standard[57].

The existing evidence for in vivo cancer associated protease activity has relied on preclinical models[41,58] and the library of proteolytically derived peptides found in human serum[40]. Analysis of the serum peptidome demonstrated tumor specific patterns of proteolytic degradation that support a diagnostically useful signature of peptide generating enzymes. In mass spectrometry approaches, significantly greater signal was observed in serum

than in plasma consistent with our observations (Supplementary Fig. 8). Thus, there appears to be a direct link between patterns of proteolytic activity and disease status. In this setting, artificial protease substrates can usefully serve as surrogate biomarkers to the serum peptidome for the detection and classification of cancer[59].

There are numerous efforts to develop additional biomarkers with utility in the detection of lung cancer. These approaches include modifications to the risk group classification[6,60] or blood based biomarkers. Biomarkers are typically related either to the inflammatory process associated with the onset of cancer[61,62] or circulating free nucleic acids (cfDNA). The cfDNA space has progressed from an analysis of mutations associated with disease[63] to patterns of methylation[64] or fragment characteristics[37]. The challenge with all these approaches is that they are fundamentally biomarker counting technologies. As such, they are limited by the tumor volume and there are very real biological limitations to the number of biomarkers that can be counted. These results compare favorably with results published for other validated blood tests designed for lung cancer detection. The validated Galleri test (Grail, Menlo Park CA) has a sensitivity for Stage I lung cancer of 22%[65]. Similarly, the EarlyCDT lung test produced an overall sensitivity of 33% in a high risk cohort study with 21% sensitivity in Stage I–II lung cancer patients[66]. Somalogic reported 79% sensitivity and 71% specificity in Stage II to IV validation samples from the Early Detection Research Network biobank[52]. Nucleix validated a 87% sensitivity and 64% specificity using a PCR based methylation assay[67]. The advantage of using an ABS is that single molecule events, being enzymes, can generate multiple signals from a biosensor resulting in amplified signal and performance.

This study has some limitations being addressed by ongoing data collection. Being designed primarily around cohort samples, the real-world performance in an all-comers LDCT screening setting is limited to a subset of the samples. Despite this, the non-lung cancer control group is drawn from age and smoking matched patients. Further, the machine learning tools have been focused on the presence or absence of disease. To draw accurate boundaries between pulmonary inflammation and cancer, more samples will have to be examined. For instance, SCLC is only represented at 1% in this cohort. These data have been restricted to a defined clinical risk population so a finding of abnormal activity may be assigned to lung cancer. However, as a platform technology, the definition of activity changes associated with disease will need to be further elaborated to use protease activity in multiple different cancers.

In summary, serum-based ABS are suitable tools to detect protease activity associated with the presence of lung cancer. To define this system, we identified targets selective for proteases most differently regulated in patients with or without lung cancer and used machine learning to distinguish normal and abnormal enzyme activity. This approach results in an assay with high performance in the context of detecting lung cancer and a cost effective and rapid turnaround. Ongoing validation studies focused on all comers screening studies and other cancers could result in a tool very well suited to population wide screening.

## Data availability
All source data for all the figures in the main manuscript is available in the Supplementary Data 1.xlsx file available online. Requests for access to the Ensemble Excel model will undergo a prompt review to ensure the request is not subject to any intellectual property or confidentiality obligations. Access to Ensemble Excel model data will be subject to a data transfer agreement. Requests to access this data set should be directed to the corresponding author.

## Code availability
The Emerge software is a service product from Liquid Biosciences, Inc. and is not available for release to the public. Liquid Biosciences Inc. is a commercial entity and their services can be invited by any interested party.

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

## Author contributions

P.W.D., S.H., N.K.V., N.O.E., T.F.H., and O.H. conceived of and designed the clinical studies. C.-M.S., R.G., S.H., P.W.D., S.H.B., and O.C.-Z. designed and manufactured the biosensors. A.S.N., N.K.V., N.O.E, D.K., T.L., B.Y., S.W.C.W., L.N., T.F.H., O.H., collected and organized clinical samples. Biosensor measurements were performed by C.-M.S., R.G., S.H. P.L., S.H., and P.W.D. analyzed the experimental data and developed the classification model. A.S.N., N.K.V., N.O.E., S.W.C.W., and L.N., reviewed clinical data. P.W.D, S.H., C.-M.S., wrote the first draft of the manuscript. All authors have reviewed and approved the final version of the manuscript.

## Competing interests

P.W.D., C.-M.S., S.H., O.C.-Z. and S.H.B. are co-inventors in a provisional patent application (WO/2022/147171A1) covering the biosensor described in this article. R.G. is an employee of Hawkeye Bio, Inc. P.L. is a co-founder and employee of Liquid Biosciences Inc. PWD is a founder at Hawkeye Bio, Inc. S.H.B. and N.O.E. are consultants for Hawkeye Bio, Inc. S.H.B, N.O.E, L.N., and O.H received research funding from Hawkeye Bio, Inc. S.W.C.W., L.N. and O.H. received unrelated research funding from NK MAX. A.S.N, N.K.V., D.K., T.L., and B.Y. have no competing interests to declare.
