## [Peer Review File · Communications Medicine]

Reviewers' comments:

Reviewer #1 (Remarks to the Author):

Title: Description of an activity-based enzyme biosensor for early stage lung cancer detection.

Manuscript ID: COMMSMED-23-0485-T

Corresponding Author: Paul W. Dempsey

In this study, the authors showed an activity based sensor (ABS) platform to measure active protease enzymes in serum samples from individuals at higher risk for lung cancer. By using a machine learning approach, the ABS panel successfully detected 90% of lung cancer patients with an overall specificity of 82%, including 90% sensitivity in Stage I disease, showing potential as a clinically useful method for early lung cancer detection and classification of the inflammatory response to cancer. This study presents an approach by utilizing a panel of various protease measurements to enhance the accuracy of early lung cancer detection through AI. However, the observed specificity of 82% raises concerns about its adequacy for disease diagnosis. The absence of comparative data with other diseases calls into question the specificity of the method towards lung cancer. Despite using multiple patient samples, several limitations in the suitability of this diagnostic method for early lung cancer detection need to be addressed. Additionally, the insufficient main data may warrant reconsideration for publication in this journal. Therefore, this paper is not suitable for publication in this journal.

Reviewer #2 (Remarks to the Author):

The authors address early detection for the cancer recognized for the highest mortality. It seems a feasible and appropriate to study the enzyme levels in serum (vs plasma) to provide (early) detection tool.

Major comments:

The authors analyze the assay performance among all cancer stages while discussing that this approach is meant for the early detection.

Already in the abstract, it becomes unclear why authors discuss machine learning approach for the Stage I disease prediction to follow up with the comment on all stages.

The performance (Suppl Table 4) of the LEAP panel using sera is even higher for more advanced stages, therefore the underlying aim becomes questionable. These results should be discussed in more depth. Important question arises: are the proposed biomarkers indeed a good criterion for the early stage as authors try to make a case of?

Moreover, authors title the article as focusing on the early stage yet in Discussion they mention 'clinically useful detection of 204 lung cancer across all stages' with specificity and sensitivity listed for stage I only. At the same time the performance is high across all stages if not higher for more

advanced stages (same suppl table).

Specific questions:

The sentence: 'This methodology evolves algorithms' makes no sense. Once corrected, it still needs to be supported by a follow up statement. Most of the readers are aware of the algorithmic nature of the ML models.

The sentence: 'The 158 Training, Selection, and Test data subsets were scrupulously segregated, to avoid information leakage between discrete steps in the modeling process' is unclear. What leak of information do authors have in mind? Although scrupulously, how exactly were the subsets segregated with patients' characteristics in mind?

Table 3. Performance. Authors should comment of the low specificity values, what affects them given the input data, and discuss how they could improve it. I.e., authors should connect the model performance results with false incidents. Especially there is a need to discuss a high False Positives group.

Authors do not provide sufficient details on the machine learning model applied and/or alternative models to test the performance. There is no information if any performance tests were applied, e.g., cross-validation, resampling.

Were the data points selected for each set resampled and what is the Selection Set? Authors could provide more information if this corresponds to validation data.

What ML model is selected and why?

What was the computational model performance in terms of convergence? Can the model be interpreted more in the article?

What stands behind a 'single mathematical' or 'logical' function? What drove the selection of this function?

What is the nature of the 'initial algorithmic models'?

What 'information theory' algorithms did the authors use and to address which question?

Reviewer #3 (Remarks to the Author):

Comment: Accepted after minor revision

The manuscript presents a new method utilizing machine learning in the protease enzyme biosensor for lung cancer diagnosis. The author claims this method could differentiate early stage (Stage I) lung cancer from healthy control. However, the clinical specificity of enzyme activity for lung cancer diagnosis is not very well presented, which greatly hinder the understanding for following data analysis and clinical conclusion.

I recommend publication in Nature communication Medicine after minor revision.

Minor Comments for consideration:

(1) Study design: The clinical cohort is not very well described. Are those pre-collected samples from other study, or specifically collected for this study? Is this study prospective or retrospective? What is the statistical power in the study? In this study, what is the enrollment criteria – are subjects in high risk group for lung cancer, or subjects already developed symptom? What is the definition of the control group? Does those control subjects also have LDCT results, or biopsy result? If the author could provide more detailed information for the study design that will help a lot in the data analysis and clinical conclusion.

(2) An illustration for the sensor fabrication will be very helpful to understand the whole procedure. In addition, sensor characterization other than DLS and zeta potential will be helpful.

Reviewer #1 (Remarks to the Author):

Title: Description of an activity-based enzyme biosensor for early stage lung cancer detection.
Manuscript ID: COMMSMED-23-0485-T
Corresponding Author: Paul W. Dempsey

In this study, the authors showed an activity based sensor (ABS) platform to measure active protease enzymes in serum samples from individuals at higher risk for lung cancer. By using a machine learning approach, the ABS panel successfully detected 90% of lung cancer patients with an overall specificity of 82%, including 90% sensitivity in Stage I disease, showing potential as a clinically useful method for early lung cancer detection and classification of the inflammatory response to cancer. This study presents an approach by utilizing a panel of various protease measurements to enhance the accuracy of early lung cancer detection through AI. However, the observed specificity of 82% raises concerns about its adequacy for disease diagnosis. The absence of comparative data with other diseases calls into question the specificity of the method towards lung cancer. Despite using multiple patient samples, several limitations in the suitability of this diagnostic method for early lung cancer detection need to be addressed. Additionally, the insufficient main data may warrant reconsideration for publication in this journal. Therefore, this paper is not suitable for publication in this journal.

We thank the reviewer for the comments. Within this paragraph, we understand three questions (underlined) the reviewer brings up to be addressed:

1. Specificity of 82% raises concerns about its adequacy for disease diagnosis.

We thank the reviewer for this question which raises an important point that we would like to clarify here and in the paper. The LEAP test is designed as a triage test to provide decision support for lung cancer screening to physicians and their patients. It is not designed as a diagnostic tool or a screening tool. In response to these helpful criticisms, we have modified our presentation to better contextualize the LEAP performance as a decision support tool.

Diagnosis of lung cancer is not achieved with a single modality. LDCT imaging, the screening standard in the US, identifies only the presence of a suspicious nodule. In the NSLT, 7,191 patients of the 26,309 LDCT participants had a positive screening result. Follow up diagnostic procedures confirmed lung cancer in only 292 participants. This represents 4.1% of the population that were screen positive¹.

In its review of evidence for LDCT screening, the USPSTF examined 10 randomized control trials which returned 80.3% Sensitivity and 76.4% Specificity mean performance². The reported LEAP performance of 90% sensitivity and 82% specificity should be interpreted in this context. We have included a discussion of this background as well as some other validated tests designed to be deployed in this setting (Page 11, lines 241-251).

“During review of the risk recommendations, the USPSTF examined 10 randomized control LDCT screening trials for accuracy and documented a mean 80.3% Sensitivity (range 59.0%-95.0%) and 76.4% Specificity (range 26.4%-99.2%)(Jonas, Reuland et al. 2021). In the context

¹ Team, N. L. S. T. R. et al. Results of Initial Low-Dose Computed Tomographic Screening for Lung Cancer. *New Engl J Medicine* **368**, 1980–1991 (2013).

² Jonas, D. E. et al. Screening for Lung Cancer With Low-Dose Computed Tomography. *JAMA* **325**, 971–987 (2021).

of informing patient selection for LDCT screening therefore, the balanced Ensemble has a performance that could serve as a decision support tool for LDCT screening. The tunable nature of the Ensemble allows consideration in other regions outside the US that may have different selection criteria and different future implementations of LDCT screening. A bias towards Sensitivity is seen in other tests. Mathios *et al.*, report a combination of clinical risk, CEA protein levels, and fragmentation profiles to detect 91% stage I/II patients at 80% specificity(Mathios, Johansen *et al.* 2021). A four biomarker panel evaluated on Prostate, Lung, Colorectal, and Ovarian (PLCO) Cancer Screening Trial samples returned a Sensitivity of 85.0% and Specificity of 71.1%(Irajizad, Fahrman *et al.* 2023)."

2. *The absence of comparative data with other diseases calls into question the specificity of the method towards lung cancer.*

These data focus on patients recruited in LDCT screening or smoking cessation settings and so are not suited to analyze the performance of other cancers. Within a population at risk for lung cancer, our approach has captured the relevant comorbidities associated with a population of older heavy smokers. We have future efforts planned to examine the LEAP profile in other cancers.

3. *several limitations in the suitability of this diagnostic method for early lung cancer detection need to be addressed.*

This manuscript describes the performance of a blood based ABS to distinguish older heavy smokers from the same population but with pathologically confirmed lung cancer. The performance is reported on an out-of-sample Test set of 250 samples. We acknowledge in our closing comment that "Ongoing validation studies focused on all comers screening studies and other cancers could result in a tool very well suited to population wide screening." However, as a technology that can add important information to the population at increased risk for lung cancer, the LEAP test compares very favorably with other validated approaches, examples of which we have included in this revision (Page 13, lines 324-330):

"These results compare favorably with results published for other validated blood tests designed for lung cancer detection. The Galleri test (Grail, Menlo Park CA) has a sensitivity for Stage I lung cancer of 22%(Klein, Richards *et al.* 2021). Similarly, the EarlyCDT lung test produced an overall sensitivity of 33% in a high risk cohort study with 21% sensitivity in Stage I-II lung cancer patients(Borg, Wen *et al.* 2021). Somalogic reported 79% sensitivity and 71% specificity in Stage II to IV validation samples from the Early Detection Research Network biobank(Mehan, Williams *et al.* 2014). Nucleix validated a 87% sensitivity and 64% specificity using a PCR based methylation assay(Gaga, Chorostowska-Wynimko *et al.* 2021).

Reviewer #2 (Remarks to the Author):

The authors address early detection for the cancer recognized for the highest mortality. It seems a feasible and appropriate to study the enzyme levels in serum (vs plasma) to provide (early) detection tool.

Major comments:

The authors analyze the assay performance among all cancer stages while discussing that this approach is meant for the early detection.

Already in the abstract, it becomes unclear why authors discuss machine learning approach for the Stage I disease prediction to follow up with the comment on all stages.

We thank the reviewer for his comments. This brings up an important point also alluded to by Reviewer 1 that speaks to the challenge of effective cancer screening.

As stated by the National Cancer Institute, “the benefit of screening derives from detecting cancer in earlier and more treatable stages, and thereby, reducing mortality from cancer”³. So the fundamental goal of screening is to identify biomarkers that are informative in the earliest asymptomatic stages of disease. That said, any biomarker that does not detect more advanced disease will fail because more advanced patients will always present for screening even though the goal is to detect disease early in a population. In the NLST, 30% of subjects with a positive finding in the LDCT arm presented with Stage III or IV lung cancer (189 of 635 positively screened cancer patients)⁴.

To the reviewers point, this tool is not focused only on early lung cancer detection, rather it is capable of identifying subjects with lung cancer at all stages, including asymptomatic early stages where treatment outcomes are favorable. We have therefore modified the title to indicated “Description of an activity-based enzyme biosensor for lung cancer detection” eliminating the reference to “early stage” (Page 1 line 2). We have similarly modified the reference to “early detection” to “detection” in the abstract (page 1 line 6) and in the body of the manuscript acknowledging that current screening paradigms include patients with all stages of disease. To clarify the benefit of a tool that can provide diagnostic information in localized disease, we have modified the abstract comment on Stage I disease to indicate

“...90% sensitivity in Stage I when disease intervention is most effective.” (Page 2 line 23).

The performance (Suppl Table 4) of the LEAP panel using sera is even higher for more advanced stages, therefore the underlying aim becomes questionable. These results should be discussed in more depth. Important question arises: are the proposed biomarkers indeed a good criterion for the early stage as authors try to make a case of?

Protease enzymes have a pivotal role to play in multiple aspects of cancer development. Manipulation of the extracellular environment is relevant to all stages of disease (see references 9 and 10). The

³ PDQ® Screening and Prevention Editorial Board. PDQ Cancer Screening Overview. Bethesda, MD: National Cancer Institute. Updated 08/23/2023. Available at: <https://www.cancer.gov/about-cancer/screening/hp-screening-overview-pdq>. Accessed 09/27/23. [PMID: 26389235]

⁴ National Lung Screening Trial Research Team; Aberle DR, Adams AM, Berg CD, Black WC, Clapp JD, Fagerstrom RM, Gareen IF, Gatsonis C, Marcus PM, Sicks JD. Reduced lung-cancer mortality with low-dose computed tomographic screening. N Engl J Med. 2011 Aug 4;365(5):395-409. doi: 10.1056/NEJMoa1102873. Epub 2011 Jun 29. PMID: 21714641; PMCID: PMC4356534.

diagnostic model was developed to distinguish “patients with or without lung cancer” (page 4, line 62). We have not investigated if these activities can distinguish different stages.

We have modified Supplemental Table 4 (now Supplemental Table 6) to include Clopper-Pearson Exact confidence intervals calculated for each stage of disease. In this setting, the balanced Ensemble has a 95% CI of 73-98% examining 29 Stage I samples in the Test set. The 95% CI for 18 Stage IV samples is 54-100%. So the Stage I confidence intervals rest within the Stage IV performance. We are not comfortable concluding that there is a stage related improvement in performance based on these data.

We have included the following elaboration in response to the reviewers point (Page 9, lines 195-199):

“... (Supplementary Table 6). The Test set demonstrated an incremental performance of 90%, 74%, 94%, and 100% Sensitivity for Stage I through IV respectively using the Ensemble. Within these subsets, the 95% CI were overlapping suggesting while LEAP maintains clinically relevant performance across all stages of disease, additional data will be needed to more clearly define performance boundaries.”

Moreover, authors title the article as focusing on the early stage yet in Discussion they mention 'clinically useful detection of 204 lung cancer across all stages' with specificity and sensitivity listed for stage I only. At the same time the performance is high across all stages if not higher for more advanced stages (same suppl table).

The reviewer is correct to point out the value as a screening support tool is to detect cancer across all stages. We eliminated the reference to early disease in the discussion of other biomarker examples in line 240. However there is value in a focus on Stage I and II detection as this is the setting where most other tools have not been able to maintain performance. We have added other validated examples that display overall sensitivity between 33% and 87% in lung cancer patients in response to question 2 from Reviewer 1 above.

That said, in a screening setting the disease stage is unknown so the reviewer is correct to recognize a focus on early stage only does not support the application. We maintain however that it is important to have “clinically useful detection across all stages” and so have modified the discussion statement (Page 11, Lines 236-237) to read now:

“In these cohorts, we observed a sera-based specificity of 82% and sensitivity of 90% across all stages of disease.”

Specific questions:

The sentence: 'This methodology evolves algorithms' makes no sense. Once corrected, it still needs to be supported by a follow up statement. Most of the readers are aware of the algorithmic nature of the ML models.

We welcome the opportunity to clarify. This sentence (Page 17, lines 440-445) was intended to convey that Emerge:

“uses predictive or classification fitness in an evolutionary setting to evolve by mutation, cross-over or migration, and then naturally select Turing machines selecting for improved algorithmic progeny with each generation. By evaluating on the order of 10^{15} algorithms, Emerge effectively replicates natural selection *in silico* using Turing machines to capture biological aspects of the relationship between the biosensor activity and the presence or absence of disease”

The sentence: 'The 158 Training, Selection, and Test data subsets were scrupulously segregated, to avoid information 159 leakage between discrete steps in the modeling process' is unclear. What leak of information do authors have in mind? Although scrupulously, how exactly were the subsets segregated with patients' characteristics in mind?

The records in the dataset were divided into three approximately equally-sized groups, labeled as Training, Selection, and Test. The patients were randomized for disease status, disease stage, trial site, sex, patient age, smoking status, and histology and so all were represented approximately equally in each group. If any record in Selection or Test were available to the evolutionary algorithm generation process, there would be information leakage because Emerge would be able to use the Selection or Training data to generate the algorithmic models. Other machine learning, statistical, or AI approaches typically divide datasets into Training and Validation sets, and no record in Validation is permitted to be used for Training. We use the same principle, except that we (a) generate a population of algorithmic models using the Training records, in an evolutionary process, then (b) use the held-out Selection records to evaluate the models *after* the evolutionary process, and automatically select a single algorithm out of the final algorithm population. Then (c) that single final algorithm is tested against the held-out Test records. The Test records are therefore fully blinded from both Training and Selection processes, not for any purposes of further selection but to estimate out-of-sample performance assuming that any future prospective records were generated in the same way (same patient selection criteria and exactly the same measurement process) as the original dataset.

We have modified the relevant paragraph to clarify this process (Page 8, Lines 171-186):

“The software is agnostic to the nature of a problem in terms of explanatory variables, dimensionality or underlying mathematical relationships. Rather, it identifies both key variables and mathematical relationships associated with outcomes of interest. The data were divided into distinct, **non-overlapping** random subsets that were sequentially processed: A Training set, a Selection set, and a Test set segregating a third of the samples **balanced for disease status, disease stage, trial site, sex, patient age, smoking status, and histology so all are represented approximately equally in each set.** Analysis of the Training set provided the initial algorithmic models **which were randomly generated using biomarker variables and mathematical or logical functions selected from a palette of 48 functions.** These models are iteratively selected and mutated *in silico* to evolve models that identify relationships that support accurate classification. The resulting calculations were then evaluated on the held-out Selection subset to select a final model **independent of the Training set.** The performance of the final Ensemble model was confirmed on the **third out-of-sample** Test set. The Training, Selection, and Test data subsets were segregated as described, to avoid information leakage between discrete steps in the modeling process. **As such, no resampling or cross-validation was performed to avoid overstatement of the prospective out-of-sample performance.**”

Table 3. Performance. Authors should comment of the low specificity values, what affects them given the input data, and discuss how they could improve it. I.e., authors should connect the model performance results with false incidents. Especially there is a need to discuss a high False Positives group.

We have added a discussion of the specificity performance in the context of lung cancer screening to the discussion section referred to in response to Reviewer 1 above. We have also added a paragraph describing modifications that might impact specificity in future development efforts (Page 11, lines 260-269).

"We note that the sensors design was restricted by expression analysis of normal and lung cancer tissue. There are classes of proteases globally expressed that have clear impact on lung cancer that could be included in future panel designs to improve the performance in other patient risk groups. The Complement fragment C4d has been shown to be diagnostically elevated in bronchial fluids and saliva(Ajona, Razquin et al. 2015). Analysis of peptidome profiles in lung cancer also detected degradation products of Complement C3 and C4 activation further implicating the inflammatory role of the Complement cascade(Villanueva, Shaffer et al. 2006). The Kallikrein (KLK) family of protease, specifically KLK6, KLK13 and KLK14, have been implicated in a preclinical model of protease detectors for lung cancer(Kirkpatrick, Warren et al. 2020). These proteases may provide additional useful targets for ABS with impact on Sensitivity or Specificity(Lenga Ma Bonda, lochmann et al. 2018)."

Furthermore, we acknowledge there is a difference between examining cohort data in the setting of lung cancer from a screening application where the incidence of lung cancer will be significantly lower. These data will be generated in future studies. Our goal here was to define the basic performance of a novel technology (Page 11, lines 252-259).

"The subjects recruited for this study used regional LDCT classification rules that will reflect the range of specificity attributable to LDCT. Without longitudinal sampling of a defined cohort, it is unclear if part of the 82% specificity observed with the LEAP assay may capture an overlap between LDCT false negative and LEAP false positive signal. Longitudinal collection of screening samples to measure LEAP sensitivity will require future clinical trials. There is also a possibility of overlap between the LEAP profile for lung cancer with additional inflammatory diseases including other cancers. However, application to a particular risk group restricts the clinical use case to lung cancer rather than a multi-cancer detection tool.

Authors do not provide sufficient details on the machine learning model applied and/or alternative models to test the performance. There is no information if any performance tests were applied, e.g., cross-validation, resampling.

The Emerge approach specifically excludes cross-validation or resampling approaches. As described above, the full dataset is divided into three approximately equal sets of data that have no overlap: Training, Selection, and Test. Training is performed on the first discrete subset of data. The evolutionary process randomly generates and then optimizes, using an evolutionary process, a population of algorithmic models. The Emerge system's evolutionary process uses tournament selection at each generation, to select half of the algorithms in the population based solely on accuracy in the Training subset of the patient records. The selected algorithms reproduce by either mutation or cross-over between two selected algorithms. The parent algorithms and these progeny survive to the next generation, until the final generation. This process optimizes model form, variables, and mathematical functions to maximize accuracy by finding optimally explanatory relationships in the training subset of patient records

Selection is performed on the second subset of data used for interim validation such that an automatic heuristic (the selection rules are proprietary) can be applied in order to select a single algorithmic model (final model) out of the population. Test is the subset of data used as a fully-blinded hold-out to allow estimate of the final model's prospective performance.

Cross-validation is irrelevant; this is a technique used by other methods in order to estimate out-of-sample performance, which we estimate instead as described above. Cross-validation is not an appropriate method in any case, because it does not lead to selection of a single model to use clinically; instead it generates multiple models on different training subsets and "validates" them on the remaining records in each case. Such techniques produce some number of models (e.g. "n"

models in the case of “n-fold cross-validation”) and average their performance as an estimate of prospective performance, but they do not provide any valid means of selecting one of the models.

As for re-sampling, our approach does not require it. We randomly divide the records into Training, Selection, and Test subsets, preserving proportions of the diagnostic outcomes. We do that only once, because doing it more than once would allow selection of the permutation that produces the “best” results, which is information leakage and therefore would overstate the estimate of prospective performance.

Were the data points selected for each set resampled and what is the Selection Set? Authors could provide more information if this corresponds to validation data.

Please see the explanation to the previous question.

What ML model is selected and why?

We refer the reviewer to the discussion above regarding the Emerge system as the selected approach. We presented a comparison of Emerge to binomial logistic regression at IASCL 2022 using a commercially sourced sample set. The Sensitivity and Specificity using Emerge was 97% and 82% as compared to 79% and 65%⁵. These data described a significant improvement in performance using Emerge which informed our selection of this approach.

What was the computational model performance in terms of convergence? Can the model be interpreted more in the article?

We understand this question to mean the settling of the dispersion of errors around the actual value, as one accepted meaning of “convergence”. There is no relevant measure of convergence other than a confidence interval calculation based on the Bernoulli distribution of the cancer/not prediction.

It is worth pointing out that the Emerge approach has no information about what the biomarkers are. The biomarkers are arbitrarily assigned names BM01 etc., without context. So interpretation of the model output must be based on the ranking of the diagnostic impact of each of the biosensors. We expand our analysis of the biomarker contribution presented in Figure 3 with the understanding that we are interpreting activity in the collected sample rather than a tissue specific measurement (Page 12, lines 276-294).

“In our analysis of the impact of each of the biosensors (Figure 3), all the sensors were selected by Emerge to have contributed positively to the diagnostic performance. Allowing for the promiscuity of protease activity^(Holt, Lim et al. 2022), the literature does support a role for many of the selected targets. Both MMP2 and its inhibitor TIMP are elevated in bronchial alveolar lavage fluid^(Cao, Xu et al. 2017) and serum^(Hoikkala, Paakko et al. 2006) from lung cancer patients. Cordes *et al.*, examined the expression of all 13 Cathepsins in the context of lung cancer and found over expression of both CTSB and CTSK were significantly associated with poorer 5-year survival^(Cordes, Bartling et al. 2009). Previous observations by Werle *et al.*, demonstrated significantly altered expression of CTSB and uPA and unfavorable prognosis associated with increased tissue activity of CTSB^(Werle, Kotzsch et al. 2004). CTSH interestingly shows little change in protein expression in tissue samples but increased detection in sera maybe reflecting increased secretion, especially in smokers^(Schweiger, Staib et al. 2000). Arginase 1 and 2 expression is increased in lung cancer tissue^(Niu, Yu et al. 2022). Arginine depletion is suggested to alter T cell responses *in*

⁵ Dempsey, P. W., Aparicio, C.-M. S., Hantula, S., Covarrubias-Zambrano, O. & Bossmann, S. H. EP01.01-010 Graphene Based Activity Sensors Detect All Stages of Lung Cancer Using an Evolutionary Machine Learning Algorithm Approach. *J Thorac Oncol* **17**, S164–S165 (2022).

vitro(Niu, Yu et al. 2022) but it is not associated with worse prognosis unless cancer associated fibroblast ARG2 expression is examined(Giatromanolaki, Harris et al. 2021) possibly consistent with the boundary region expression of Arginase activity in a preclinical model(Malalasekera, Wang et al. 2017). Both MMP7 and MMP12 protein expression is increased in both tissue and matched serum samples(Mehan, Williams et al. 2014). MMP9 has been shown to be both over-expressed(Blanco-Prieto, Barcia-Castro et al. 2017) and show elevated tissue activity in NSCLC tissue(El-Badrawy, Yousef et al. 2014). Conversely, activity based sensors showed elevated MMP1 and MMP2 activity but not MMP9 activity relative to control tissue samples emphasizing the necessity of repeatable standardized measurement tools(Yoneyama, Gorry et al. 2018)."

What stands behind a 'single mathematical' or 'logical' function? What drove the selection of this function?

The initial mathematical or logical functions are single math or logic operations, like addition, multiplication, IF, cosine, etc. Emerge starts the evolution process with a palette of up to 48 different math and logic functions that may be included in the initial-random digital genome of every one of the many algorithms created immediately prior to the evolutionary process beginning. During the evolutionary process, no functions are selected or deselected. Instead, algorithms either survive or are removed from the population based on their fitness (a composite of sensitivity and specificity) relative to other algorithms in the population. To the extent an algorithm stays in the population, then the math/logic functions it uses stay in the population. To the extent that a math/logic function isn't as useful as others, the algorithms that use it will tend to fall out of the population. As generations of evolution progress, math/logic functions that are most useful will tend to be used by algorithms that perform well, and math/logic functions that are less useful will tend to be used in a smaller and smaller proportion of the population of surviving algorithms, until such math/logic functions disappear from the population entirely. There is therefore no direct "selection" or "deselection" of any function; each function's contribution (like any variable's contribution) matters only in the context of how it is used with the other functions in each algorithm. We have edited the methods section description of Emerge to include the following clarification (Page 17, lines 440-445):

"This methodology uses predictive or classification fitness in an evolutionary setting to mutate and naturally select Turing machines selecting for improved algorithmic progeny with each generation. By evaluating on the order of 10^{15} algorithms, Emerge effectively replicates natural selection *in silico* using Turing machines to capture biological aspects of the relationship between the biosensor activity and the presence or absence of disease."

What is the nature of the 'initial algorithmic models'?

Both initial and subsequent (i.e. during and after evolution) algorithmic models are comprised of a sequence of mathematical or logical functions ("instructions"), each of which applies a math or logic function to one or two operands (variables or constants). Each initial algorithmic model is randomly created from the raw materials of a palette of math and logic functions, the available variables (data like biomarkers variables), and a pool of constants (numbers). After the initial random creation, evolution begins. As described above, in every generation each algorithm's accuracy is compared to other algorithms in the same generations population. The algorithm with better accuracy (again, a composite of sensitivity and specificity) survives, and the other one is removed from the population. Then the surviving algorithm reproduces either asexually (a clone is made, which then undergoes a mutation in the form of a change to one instruction), or sexually (two surviving algorithms create two children from segments of each parent algorithm). The parents survive also.

What 'information theory' algorithms did the authors use and to address which question?

There are no "information theory" algorithms. Information theoretic measures of the algorithms' performance is used during Selection, but the specifics are trade secrets.

Reviewer #3 (Remarks to the Author):

Comment: Accepted after minor revision

The manuscript presents a new method utilizing machine learning in the protease enzyme biosensor for lung cancer diagnosis. The author claims this method could differentiate early stage (Stage I) lung cancer from healthy control. However, the clinical specificity of enzyme activity for lung cancer diagnosis is not very well presented, which greatly hinder the understanding for following data analysis and clinical conclusion.

We thank the reviewer for his comments and positive review. We have modified the discussion of specificity both in terms of the technology and the clinical need to address the question of specificity as discussed in response to the previous reviews.

We acknowledge that a diagnostic tool designed to improve the rate of screening for lung cancer has to perform across all stages of cancer not just early stage disease. However, the positive impact of improvements in screening is seen when more patients are detected in early curative stages. This was the rationale behind our developmental focus on early stage performance. However, recognizing the need for correct classification of cancer across all stages we have modified our language to highlight the performance in early stages but broadly present classification performance that is relevant to the at risk population which present with all stages of cancer.

I recommend publication in Nature communication Medicine after minor revision.

Minor Comments for consideration:

(1) Study design: The clinical cohort is not very well described. Are those pre-collected samples from other study, or specifically collected for this study? Is this study prospective or retrospective? What is the statistical power in the study? In this study, what is the enrollment criteria – are subjects in high risk group for lung cancer, or subjects already developed symptom? What is the definition of the control group? Does those control subjects also have LDCT results, or biopsy result? If the author could provide more detailed information for the study design that will help a lot in the data analysis and clinical conclusion.

We have edited the brief description of the patient selection criteria in the paper to help provide better clarity. The Reporting Summary also contains an outline of the inclusion and exclusion criteria applied at each site. In short, all patients were sampled in a prospective manner. Subjects in Cohort 2 all had a recent chest CT to confirm negative status in addition to any necessary follow up procedures (such as biopsy or repeat imaging as determined by their provider) to define any indeterminate nodule observations.

“Clinical Samples: Samples were collected at two sites (HEB.115 and HEB.121) in prospective cohorts from subjects who were defined by the USPSTF criteria as at increased risk for lung cancer based on age (between 50 and 80 years of age) and smoking history (current smokers with greater than 20 pack years or cessation within the last 15 years). Subjects without lung cancer underwent chest CT screening and necessary follow up review to confirm negative status. Patients with lung cancer all had pathologically confirmed disease and were treatment naïve. A third site (HEB.130) collected prospective samples from donors identified clinically as at risk for lung cancer so included some number outside the USPSTF age and smoking limitations. Subjects at HEB.130 were followed to diagnostic resolution as negative for lung cancer by chest CT scans and

necessary follow up to confirm the absence of disease. Patients positive for lung cancer all had disease pathologically confirmed (Table 2).”

Statistical Methodology

As outlined in the nr-reporting-summary submitted with this manuscript:

“For our study, we aimed to achieve a 95% confidence interval with the probability of type I error equal to $\alpha=0.05$. For our study, we determined that a maximum marginal error of 5% was acceptable given the analytic targets of this study ($d = 0.05$). When the true status or condition is known before or during evaluation, the prevalence may effectively be controlled. In this study we used 25% as the prevalence. The estimates for sensitivity (84%) and specificity (93%) were informed by a development study performed at Hawkeye Bio using a total of 351 commercial samples.

Using these values in a power analysis, we estimated the minimal sample size required for this study was 400 participants in order to estimate the specificity. The target prevalence of disease for this study where the true positive population is known, is controlled by sample selection and set at 25%.

The study comprised 450 unique patient samples that included 133 serum samples from patients with pathologically confirmed lung cancer and 317 matched controls as described above. A subset of 150 samples selected to reflect the same site and disease distribution were selected for repeated evaluation to determine repeatability. A total of 750 samples were therefore evaluated.”

Target enrollment for this study therefore was:

- Total of 400 subjects
- 100 in Cohort I, the lung cancer cohort
- 300 in Cohort II, the control cohort

Primary Endpoint	Calculation of Needed Sample Size	Values for Sample Size Formula				
Primary Outcome Measure	Sample Size Formula	Sample Size Estimate	Critical Value ($Z_{\alpha/2}$)	LEAP Expected Value (%)	Precision (d) (%)	Population Prevalence (%)
Sensitivity	$n_{se}=(Z_{\alpha/2}^2 \times Se \times (1-Se))/(d^2 \times Prev)$	400	1.96	93	5.00	25.00
Specificity	$n_{sp}=(Z_{\alpha/2}^2 \times Sp \times (1-Sp))/(d^2 \times (1-Prev))$	275	1.96	84	5.00	25.00

(2) An illustration for the sensor fabrication will be very helpful to understand the whole procedure. In addition, sensor characterization other than DLS and zeta potential will be helpful.

We direct the reviewer to Figure 1 for a graphical illustration of the assembly and activation mechanism for the graphene based biosensors. We have modified the figure to clarify the steps used in the manufacture of biosensors and described in the methods section.

To further characterize the sensors, we have included data in the manuscript and in supplementary data to describe the following aspects.

1. Raman analysis of the core particle graphene is presented for 4 different lots of starting graphene material. The spectral analysis is consistent with multilayered graphene material, not graphite. All lots present with similar profiles. (Figure S1).

“The Raman spectra of the core particle show characteristic graphene features with D, G, and 2D peaks centered at 1340, 1575, and 2680 cm⁻¹ respectively (Figure S1).”

2. Transmission electron micrographs were examined of all 18 sensors. The number of layers present in the particles was counted on 79 different edge sites clearly identified in the electron micrographs to confirm the average number of layers in the fractals. A table describing the layer numbers calculations is included in the file Supplementary Table 1. We include a representative example of the transmission electron micrographs (Figure S2).

“The deagglomerated few-layer graphene ($n=7.2 \pm 2.3$ layers, (Supplementary Table 1 and Figure S2) was surface-carboxylated and then coated with a water soluble polyethyleneimine (PEI) skin.”

3. A Spectradyne nCS1 platform was used to measure the size distribution of the BM17 biosensor. This platform indicated the average particle size fell between 80 and 300nm and gave calculated average sizes of 87nm and 212nm using two overlapping microfluidic cartridges. Size estimation by the Spectradyne instrument does not rely on spherical approximations of particles. These size estimates are in close agreement with DLS measurements. (Figure S3)

“The particle size was also measured using a microfluidic resistive pulse sensing Spectradyne nCS1 platform. In agreement with the DLS observations, the weighted average of the biosensor particles measured between 87 and 212nm in diameter (Figure S3).”

4. Elemental analysis of the biosensors was conducted on different batches of core particle during the assembly of graphene to carboxygraphene to polyethyleneimine functionalized graphene.(Data are presented in Supplementary Table 3).

“Furthermore, elemental analysis of the biosensors during fabrication showed incremental inclusion of Hydrogen and Oxygen after carboxylation of the graphene backbone. Addition of Nitrogen was observed only after addition of the PEI skin (Supplementary Table 3).”

Reviewers' comments:

Reviewer #1 (Remarks to the Author):

I checked the previous revision comments that I provided.
I think the authors responded well regarding all comments with reasonable descriptions.
In this case, I agree that this revised version of the manuscript could be accepted for publication.

Reviewer #2 (Remarks to the Author):

While several comments have been appropriately addressed, there remains a lack of transparency regarding the mathematical aspects of certain operations behind the Emerge platform. This is concerning, especially in an age of shared data and fair, explainable AI and reproducible data/model sharing. It's important to acknowledge that mathematical principles underlie every algorithm and allow the model to be reproducible. Readers, especially data scientist, expect the complete information about the algorithms and conditions/environments used.

As a data scientist, researcher, and educator, I request the comprehensive information about the calculations used (originating from the Emerge platform), i.e., details of the model to be shared with audience. The authors' response related to Information Theory, which states, "There are no "information theory" algorithms. Information theoretic measures of the algorithms' performance is used during Selection, but the specifics are trade secrets" is unacceptable, mathematical principles that underly every model or specific metric are universal and public. This statement does not appear to meet the standards of fair AI, reproducibility, or a reputable journal.

Reviewer #4 (Remarks to the Author):

The authors have responded appropriately to the previous reviewer's concerns. Their main conclusions of this preliminary communication are mostly justified by the data presented.

A machine learning approach was applied to build algorithms that detected 90% of cancer patients overall with a specificity of 82% including 90% sensitivity in Stage I when disease intervention is most effective and detection more challenging. This approach has promise as a clinically useful platform for the classification of the inflammatory response to cancer.

The small size of the cohort study means that the work would need to be replicated in a prospective study. In view of the rapid advances being made in the management of lung cancer a trial might be preferable to observational data to determine whether the claim for clinical utility is borne out in practice.

Reviewer #1 (Remarks to the Author):

I checked the previous revision comments that I provided. I think the authors responded well regarding all comments with reasonable descriptions. In this case, I agree that this revised version of the manuscript could be accepted for publication.

We thank the reviewer for his helpful comments. They certainly improved the manuscript.

Reviewer #2 (Remarks to the Author):

While several comments have been appropriately addressed, there remains a lack of transparency regarding the mathematical aspects of certain operations behind the Emerge platform. This is concerning, especially in an age of shared data and fair, explainable AI and reproducible data/model sharing. It's important to acknowledge that mathematical principles underlie every algorithm and allow the model to be reproducible. Readers, especially data scientist, expect the complete information about the algorithms and conditions/environments used.

As a data scientist, researcher, and educator, I request the comprehensive information about the calculations used (originating from the Emerge platform), i.e., details of the model to be shared with audience. The authors' response related to Information Theory, which states, "There are no "information theory" algorithms. Information theoretic measures of the algorithms' performance is used during Selection, but the specifics are trade secrets" is unacceptable, mathematical principles that underly every model or specific metric are universal and public. This statement does not appear to meet the standards of fair AI, reproducibility, or a reputable journal.

It is worth starting by stipulating that the paper focuses on a novel technology designed to detect chemical differences between patients with and without lung cancer. In the assembly of the technology, several commercial entities are used to build the product all of which we describe in great detail including the graphene core particle, peptide TCPP assembly and of course the generation of a workable algorithm to interrogate samples. Fundamentally the validation of the algorithm is in the clinical data presented and shared, not the software used to generate the algorithm.

Reviewer 2 resists phrases on the "proprietary" nature of the "information theory" used in Emerge in asking for release of the software. The software is a service product from Liquid Biosciences, Inc and is not available for release to the public. We previously stated in our rebuttal letter "There are no "information theory" algorithms". We have modified our manuscript in the following manner.

- 1) Modification in the text: Lines 402-405 discuss the transition from Training to Selection data sets.

Lines 402-405:

- vi) *After $w + gS$ generations, the remaining algorithms are evaluated on the selection data subset with respect to both fitness and reliability, where reliability measures consistency of*

fitness between training and selection data. A single algorithm is computationally selected based on these two measures as well as proprietary information theoretic measures.

To clarify the process and remove any confusion about information theory, we have modified this to:

vi) After $w + gS$ generations, the remaining algorithms are evaluated on the Selection data subset with respect to both accuracy and reliability. Accuracy is reflected by a combination of sensitivity and specificity as measured in the Selection subset. Reliability measures consistency of this Accuracy between training and selection data subsets. A single algorithm is automatically selected based on a combination of these two measures. The accuracy of this algorithm is then measured in the out-of-sample Test data subset, not for any further selection but to provide an estimate of performance on future prospective samples from patients under the same assay process, and under approximately the same patient selection criteria.

- 2) Our submission will include an invitation for readers to apply to the authors for access to the algorithm as applied to the data in the paper. This would involve sharing the Excel spreadsheet containing the raw concentration data and the calculations used in the Ensemble model. These requests will be considered individually and we will take the time to help such readers to navigate the calculations. To this end, the data availability comments from the Reporting Summary modified as below are included in the manuscript now before the references in a new “Data availability and Code Access” statement:

“Source data for all the figures in the main manuscript are available in the Supplementary Data 1.xlsx file available online. Any addition data requests will undergo a prompt review to ensure the request is not subject to any intellectual property or confidentially obligations. Any released data including the Ensemble Excel model will be subject to a data transfer agreement. Requests to access the data sets should be directed to the corresponding author.”

- 3) Liquid Biosciences Inc., is a commercial entity, like all the other providers we cite in the paper. Their services can be invited by any interested party.

Reviewer #4 (Remarks to the Author):

The authors have responded appropriately to the previous reviewer's concerns. Their main conclusions of this preliminary communication are mostly justified by the data presented.

A machine learning approach was applied to build algorithms that detected 90% of cancer patients overall with a specificity of 82% including 90% sensitivity in Stage I when disease intervention is most effective and detection more challenging. This approach has promise as a clinically useful platform for the classification of the inflammatory response to cancer.

The small size of the cohort study means that the work would need to be replicated in a prospective study. in view of the rapid advances being made in the management of lung

cancer a trial might be preferable to observational data to determine whether the claim for clinical utility is borne out in practice.

We thank Reviewer 4 for stepping to complete the review of this manuscript. We do agree that additional prospective studies will be important. Indeed we close the manuscript with the comment:

“Ongoing validation studies focused on all comers screening studies and other cancers could result in a tool very well suited to population wide screening. “

REVIEWERS' COMMENTS:

Reviewer #2 (Remarks to the Author):

I accept authors' responses and the manuscript for publication. Although I still believe the authors should go into more details, they have made the effort of providing more explanation to the data sets, and importantly will provide the invitation to review the algorithms and data sheet for the interested data scientists.